# Quantifying the thermodynamics of protein unfolding using 2D NMR spectroscopy

Rita Puglisi [1], Oliver Brylski[2], Caterina Alfano [3], Stephen R. Martin[4], Annalisa Pastore [1] & Piero A. Temussi [1,5 ✉]

A topic that has attracted considerable interest in recent years is the possibility to perform thermodynamic studies of proteins directly in-cell or in complex environments which mimic the cellular interior. Nuclear magnetic resonance (NMR) could be an attractive technique for these studies but its applicability has so far been limited by technical issues. Here, we demonstrate that 2D NMR methods can be successfully applied to measure thermodynamic parameters provided that a suitable choice of the residues used for the calculation is made. We propose a new parameter, named RAD, which reflects the level of protection of a specific amide proton in the protein core and can guide through the selection of the resonances. We also suggest a way to calibrate the volumes to become independent of technical limitations. The methodology we propose leads to stability curves comparable to that calculated from CD data and provides a new tool for thermodynamic measurements in complex environments.

[1] UK-DRI at the Wohl Institute of King's College London, 5 Cutcombe Road, SE59RT London, UK. [2] Institute of Physical and Theoretical Chemistry, Technische Universität Braunschweig, Braunschweig, Germany. [3] Fondazione Ri.Med, 90133 Palermo, Italy. [4] The Crick Institute, 1 Midland Road, London NW1 1ST, UK. [5] Dipartimento di Scienze Chimiche, Universita' di Napoli Federico II, Napoli, Italy. ✉email: temussi@unina.it

Protein stability, which is the capability of a protein to retain its native conformation under stress conditions, remains a major theme of molecular biology. It has, however, become possible only relatively recently to attempt observation of protein stability directly in their natural in-cell environments[1]. These studies have allowed us to probe the relative and individual role of crowding and confinement, two factors that play a determinant role in protein stability. A technique of primary importance in these studies is time-resolved fluorescence microscopy[2–4], which is based on the attachment of a fluorescence probe to a protein that can then be differentially observed directly in the mesh of other cellular components[2–4]. This approach has, for instance, recently made possible to obtain valuable time-resolved dynamics and stability data from endogenously expressed proteins in different tissues of a living multicellular organism[2–4]. These exciting successes call for the possibility to increase the range of techniques able to study in-cell protein thermodynamics, an application, which is however outside the current range of applications of fluorescence microscopy. Among the techniques customarily used for in vitro protein stability studies, neither circular dichroism (CD) nor differential scanning calorimetry (DSC) can be used for the purpose since they could not intrinsically distinguish a specific protein within a crowded complex environment as it is the intracellular one. Indeed, CD, a technique well suitable to study isolated proteins, can be completely inappropriate with media that absorb at the same wavelengths populated by the dichroism bands of the protein of interest. Likewise, DSC cannot be used for mixtures.

Nuclear magnetic resonance (NMR) is the technique that could mostly resemble fluorescence microscopy in that it allows, by easy and flexible labelling of specific components, to follow protein stability directly in-cell. NMR has the additional advantage not to need any tag addition and to report on protein stability at the single residue level. Indeed, recent studies have provided encouraging results in support of the feasibility of this approach[5–7]: protein unfolding can easily be monitored by one-dimensional (1D) NMR by following resonances of residues spatially close to aromatic rings (ring-current resonances) and thus buried in the hydrophobic core at different temperatures. We previously showed, for instance, that the stability curve calculated from 1D NMR data coincided with that calculated from CD data[8]. More difficult, however, is the application of multidimensional NMR that is nevertheless essential to achieve the resolution necessary to study even small proteins. In this application, it remains essential to evaluate whether the choice of the residue(s) used to follow the transition is representative of the process of unfolding and what is the best choice. Indeed, we tried in the past to use two-dimensional (2D) NMR in crowded and confined environments, but encountered difficulties, particularly when trying to compare our results with CD data[9,10]. There are examples in the literature of the use of single easily identifiable peaks, such as the resonance if the C-terminal amide, to measure thermodynamic parameters of unfolding in-cell[5–7] but the general applicability of this approach remains unclear.

A crucial step in using NMR to measure unfolding is the selection of suitable resonances. The main problems are linked to the nature of the resonances and to their position inside the protein. A common practice is to use the amide groups in $^{15}$N-labelled samples. In principle, all NH resonances could be used to monitor the unfolding process but in practice there are several limitations. Amide protons, also when only partially exposed to water, undergo exchange with the water and this may influence the results in several ways. The second problem, i.e., the position of the NH group with respect to the protein surface, is only partially related to proton exchange. Even more important is the relationship with the mechanism of unfolding. When

monitoring unfolding of a well structure single domain protein with other spectroscopic techniques, e.g., CD spectroscopy, one observes intensity changes related to the whole secondary structure elements (helices and/or beta sheets) whereas the changes of NMR volumes reflect changes of single atoms and indirectly their location on the protein. The position of the atom in the protein architecture may reflect cooperative unfolding only if the atom is well inside the hydrophobic core. On the contrary, if the atom is positioned on external structural elements, volume changes may reflect peripheral motions.

Here, we demonstrate that the choice of the reporter is crucial for the reliable measurement of protein stability and that proper signal averaging is essential. We used for our measurements Yfh1, a small yeast protein that is an ideal model system to determine the full stability curve: this protein undergoes, in addition to heat denaturation, also cold denaturation above water freezing[8] and thus belongs to the increasing number of natural marginally stable proteins[11]. This allows recording the complete stability curve and a more careful determination of the thermodynamic parameters.

## Results

**Validation of internal standard**. We thus set to calculate the thermodynamic parameters using the volumes of the Yfh1 amides from $^{15}$N HSQC (heteronuclear single quantum coherence) spectra measured at different temperatures in the range 5–40 °C. We first faced the problem of the potential lack of linearity between populations and peak intensities in 2D NMR: the applicability of 2D NMR to follow the thermal behaviour of peak resonances depends in fact on the relationship between peak intensity (or volume) and populations. This is not linear due to resonance-specific signal attenuation during the coherence transfer periods as the result of relaxation, imperfect pulses, and mismatch of the INEPT delay with specific J-couplings, an aspect that has caused several concerns[12].

We reasoned that, if we could compare peak areas to an internal inert standard whose peak undergoes the same intensity attenuations as a function of temperature, except that due to protein unfolding, the ratio of peak intensities/volumes to that of the internal standard should faithfully monitor population changes originated from unfolding[12]. We selected for the purpose CyaY, a bacterial orthologue of Yfh1, as an internal reference that would experience all the bottlenecks of 2D NMR but should not be influenced by unfolding in the same range of temperature as Yfh1: the two proteins have the same fold (Fig. 1a) and similar relaxation parameters[13] but CyaY is still folded at temperatures at which Yfh1 is unfolded (53 versus 35 °C and no transition at low temperature for CyaY)[14,15]. Among the 104 $^{15}$N HSQC backbone resonances of the CyaY amides, we selected the resonance of Tyr69 since it is isolated and well distinct from those of Yfh1. The amide of this residue is buried and involved in a H-bond with the oxygen of Leu62, thus also experiencing little exchange with the solvent. We recorded 2D NMR spectra of a mixture of uniformly $^{15}$N-labelled Yfh1 and Tyr69 selectively $^{15}$N-labelled CyaY (Fig. 1b) and corrected the volumes for the internal standard (see "Methods").

To calculate the thermodynamic parameters from CD or NMR data, it is generally assumed that the unfolding transition is a two-state process between folded (F) and unfolded (U) states and that the difference in heat capacity between the two forms ($\Delta C_p$) is temperature independent. When these assumptions are matched, the populations of the two forms at a given temperature, $f_F(T)$ and $f_U(T)$, are a function of the difference in free energy, $\Delta G^o(T)$, which is given by a modified Gibbs–Helmholtz equation. The curves corresponding to this equation, referred to as stability

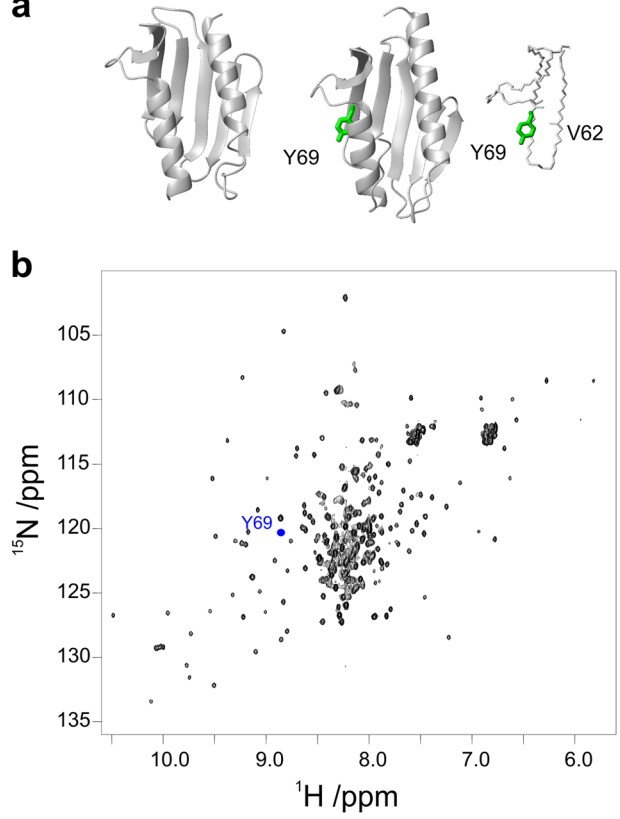

**Fig. 1 Choosing an internal reference for Yfh1 spectra. a** Ribbon representations of Yfh1 (left) and of CyaY (middle) with a detail of the hydrogen bond between Y69 NH and L62 CO (right). **b** $^{15}$N-$^1$H HSQC spectrum of Yfh1. The blue dot is placed at the coordinates of the Tyr69 amide resonance of CyaY. The models were generated by MOLMOL[34].

curves, were introduced by Becktel and Schellman[16] for a complete characterisation of the thermodynamic parameters. The procedure involves first determination of $T_m$, $\Delta H_m$ and $\Delta C_p$ using a non-linear fit of spectroscopic data and errors have been estimated from the fitting with a 95% confidence interval. These are the standard errors as reported by the fitting programme (Levenberg–Marquardt algorithm). The remaining parameters for low temperature unfolding ($T_c$ and $\Delta H_c$) can be simply read from the stability curve. We have widely verified the applicability of this equation for Yfh1 under several different solution conditions using CD and/or 1D NMR data[8–10,17–19].

Volumes of non-overlapping residues were corrected with respect to the internal standard and transformed into relative populations of folded Yfh1 assuming that, as previously proven[8], at room temperature unfolded forms are in equilibrium with a 70% population of folded Yfh1 (the exact value depends on environmental conditions such as pH, salt, buffer, etc.) as typical for marginally stable proteins[8,11,20]. Comparison of the stability curves obtained for uncorrected and corrected volumes of the Yfh1 amide resonances showed that correction does not significantly alter the results for this protein and the selected reference resonance (Fig. 2).

**Choice of resonances**. We then faced the crucial step of selecting representative resonances. When we used individual residues arbitrarily chosen among the non-overlapping ones, we observed that different residues provided highly variable thermodynamic data (Table 1 and Fig. 3a, b). The stability curves also strongly differed among each other and were significantly different from

the curve calculated from CD data used as an independent validation. Choice, for instance, of the C-terminal residue (Q174) led to parameters quite different from those obtained by CD. We reasoned that this outcome could be caused by our choice of residues far from the hydrophobic core and thus little affected by unfolding. The problem is specific for multi-dimensional NMR data while it is circumvented in 1D NMR when exploiting ring-current shifted peaks which, by their very nature, stem from residues of the hydrophobic core[8].

A simple way to select the NH resonances from residues as close as possible to the hydrophobic core can be to rely on the solvent accessible surface area (SASA) of the corresponding residue. Customarily, this value is calculated by software such as the most popular DSSP[21]. In this programme, when the SASA score lies below 50 Å$^2$, the whole residue is considered as buried. We applied this criterion to several test residues, but the results were contradictory. The approach may be misleading because the programme calculates the accessible area of the whole residue. It is well possible that the majority of the residue atoms (generally those of the side chain) are buried inside the protein whereas the amide group could be at least partially exposed. In Yfh1, this situation is aptly represented for instance by Met109, located in one of the beta strands. Its SASA is zero, reflecting the fact that the whole bulky side chain faces the interior of the protein, but the NH group points outward and may be at least partially accessible to water molecules (Supplementary Fig. 1 and Supplementary Table 1). Accordingly, intensity changes of these resonances may not reflect unfolding, even if all the side chain of the residue was completely buried. We thus looked for other criteria that could identify in a reproducible and objective way truly buried amide groups as distinct from amides of buried residues.

We looked for accessibility programmes that could yield information on single atoms. The algorithm SADIC[22] calculates intersections between the molecular volume and spheres centred on the atoms whose depth is quantified. The results for amide groups yield to a good approximation how deeply the amide group lies within the protein, i.e., below the idealised surface. SADIC thus measures the distance of an atom from the protein surface. The lower the value of the Depth parameter ($D$), the more buried the residue: residues with a $D$-value lower than 0.50 are considered buried. The programme PopS[23] yields instead relative accessibility (RA) at atomic level: Q (SASA) is the ratio between the exposed surface of a nitrogen atom with respect to the SASA of the entire residue (Supplementary Table 1).

We decided that a reasonable approach could be to calculate composite probabilities combining both the SADIC[22] and PopS[23] software. We thus defined the quantity RAD as ($D$ x RA x 100) (Supplementary Table 1), which combines depth and exposure. According to this criterion, most of the exposed residues had RAD values for the amide nitrogens considerably higher than 0.5 and were excluded from the analysis. The curves obtained for individual resonances using RAD values between 0.5 and 0.1 had a lower spread and a much better agreement with the CD curve (data not shown).

We then averaged, instead of taking individual resonances, all the non-overlapping residues with RAD < 0.5 or 0.1. We found this way a much better agreement with the CD data (Fig. 3c). For comparison, we found the curves obtained by averaging over residues in bona fide secondary structure tracts or selected for residues with RAD values <0.5 to be marginally worse. The results were almost optimal for RAD values <0.1 both in terms of cold denaturation temperature ($T_c$) and the temperature of zero entropy and at the maximum of the curve ($T_S$) (Fig. 3d). The agreement between the corresponding thermodynamic

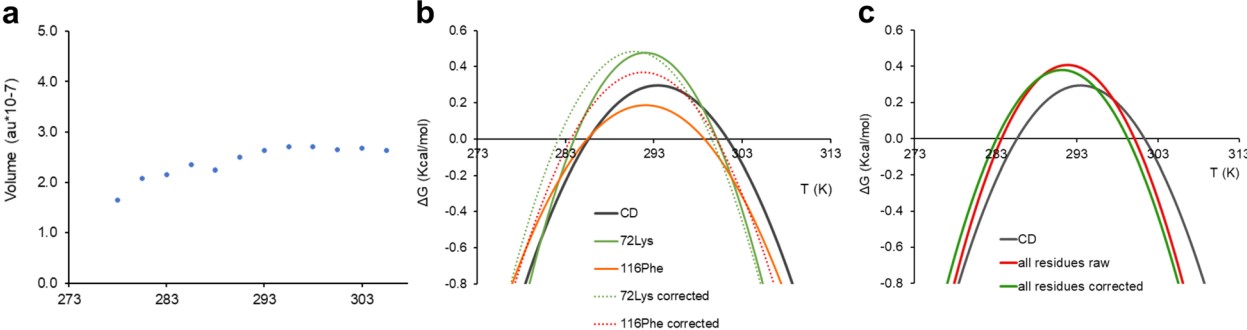

**Fig. 2 Effects of internal referencing. a** Temperature dependence of relative volumes of CyaY Tyr69 NH. **b** Stability curves from CD (grey line) and from two individual NMR resonances (Lys72 and Phe116). Corresponding dashed curves of the same colour were calculated from corrected volumes. **c** Comparison of corrected and non-corrected curves averaged over residues in secondary structure elements.

| Table 1 Comparison of thermodynamic parameters derived from CD and the different selections from NMR resonances. | | | | | |
|---|---|---|---|---|---|
| | ΔH (kcal/mol) | ΔS (kcal/mol) | ΔC_p (kcal/mol K) | $T_m$ (K) | $T_c$ (K) |
| CD | 22.5 ± 2.1 | 0.075 ± 0.007 | 2.8 ± 0.1 | 301.3 ± 0.4 | 285.6 ± 0.3 |
| Asn154 | 27.9 ± 3.6 | 0.094 ± 0.010 | 3.1 ± 0.5 | 295.4 ± 0.8 | 277.7 ± 0.2 |
| Phe116 | 26.6 ± 3.4 | 0.089 ± 0.010 | 3.2 ± 0.6 | 300.1 ± 1.3 | 283.6 ± 0.2 |
| Leu91 | 34.1 ± 3.9 | 0.114 ± 0.010 | 4.2 ± 0.4 | 300.3 ± 0.5 | 284.4 ± 0.2 |
| Lys72 | 33.1 ± 5.7 | 0.111 ± 0.020 | 3.7 ± 0.7 | 299.5 ± 0.5 | 282.1 ± 0.2 |
| Gln174 | 21.8 ± 1.2 | 0.073 ± 0.004 | 2.3 ± 0.2 | 297.7 ± 0.3 | 278.8 ± 0.2 |
| All uncorrected | 29.7 ± 1.0 | 0.099 ± 0.004 | 3.6 ± 0.1 | 300.1 ± 0.2 | 283.7 ± 0.2 |
| All corrected | 27.9 ± 1.2 | 0.093 ± 0.004 | 3.4 ± 0.1 | 299.2 ± 0.2 | 283.1 ± 0.2 |
| RAD < 0.5 | 28.9 ± 1.0 | 0.096 ± 0.003 | 3.4 ± 0.1 | 299.9 ± 0.1 | 283.3 ± 0.2 |
| RAD < 0.1 | 32.9 ± 1.2 | 0.110 ± 0.004 | 4.1 ± 0.2 | 300.4 ± 0.1 | 284.8 ± 0.2 |
| Secondary structures | 29.2 ± 0.9 | 0.097 ± 0.003 | 3.4 ± 0.1 | 300.3 ± 0.2 | 283.6 ± 0.2 |

parameters was similarly excellent within experimental error (Table 1).

The slightly worse (but anyway within 2–3 °C) agreement of $T_m$, which seems to be independent of the choice of the selection, could be explained first by the fact that the CD measurements are carried under conditions of quasi-equilibrium since the temperature increment of 1 °C/min is probably insufficient to reach full equilibrium. The effect was more marked at high temperature. NMR measurements are instead in full thermal equilibrium by definition because of the time needed to set up each experiment. Secondly, the maximal temperature reached without complete overlap of the HSQC peaks due to unfolding was 40 °C, a value close to $T_m$. This temperature does not however correspond to a completely unfolded state possibly yielding to larger fitting errors.

## Discussion
The results obtained led us to draw two important conclusions. First, we observed that the technical issues due to the non-linear relationship between peak intensity (or volume) and populations[12] are marginal in the case of Yfh1. For a different protein this may not turn true and would need ad hoc validation. We showed anyway that the use of an internal standard could provide an easy way to correct the data against possible non-linear behaviour of the volumes as a function of the temperature. Besides our specific choice, it would be easy, more in general, to find standards that retain their folding state over the whole temperature range explored, such as cystatins or the Gcn4 leucine zippers, that retain their three-dimensional structure even above 90 °C[24,25]. It is also in order to remember that, while we arbitrarily chose here Tyr69 to establish the feasibility of the method, more than one reference peak could be chosen to strengthen the conclusions in a more focused study.

A second crucial conclusion from our data is that it is possible to reliably use 2D NMR spectroscopy to monitor the unfolding of a protein with confidence and accuracy provided that the data are averaged over residues directly involved in the process of unfolding. The holy grail of protein stability is to assess thermodynamic parameters directly in-cell. So far, this information has been accessible mainly by fluorescence spectroscopy, but NMR could, in principle, be a valuable alternative. It was however crucial to check whether conclusions extracted from specific residues are equally reliable. This theme has repeatedly been debated also for other techniques such as tryptophan fluorescence spectroscopy. In this technique, the fluorescence of a specific residue is taken as representative of the unfolding transition also when the residue is exposed and far from the scene of unfolding.

We found that the choice of the residues whose behaviour is followed does matter. Residues far from the hydrophobic core and in flexible positions yield curves very different from each other raising doubts on the reliability and reproducibility of the analysis. This differential behaviour could be interesting per se but tells us relatively little about the process of unfolding if we assume a cooperative model of unfolding. We solved the problem by selecting single residues according to the RAD parameter and averaging the NMR signal over all residues within the RAD criterion. These choices yielded stability curves in impressive agreement with the CD curve, comforting the soundness of our choice. It should be noticed that residue averaging directly reproduces with NMR data the cooperative all-or-none process that we observe by other techniques such as CD or differential scanning calorimetry, which are techniques traditionally used for determining protein stability. It is also somewhat equivalent to what is done in 1D NMR when

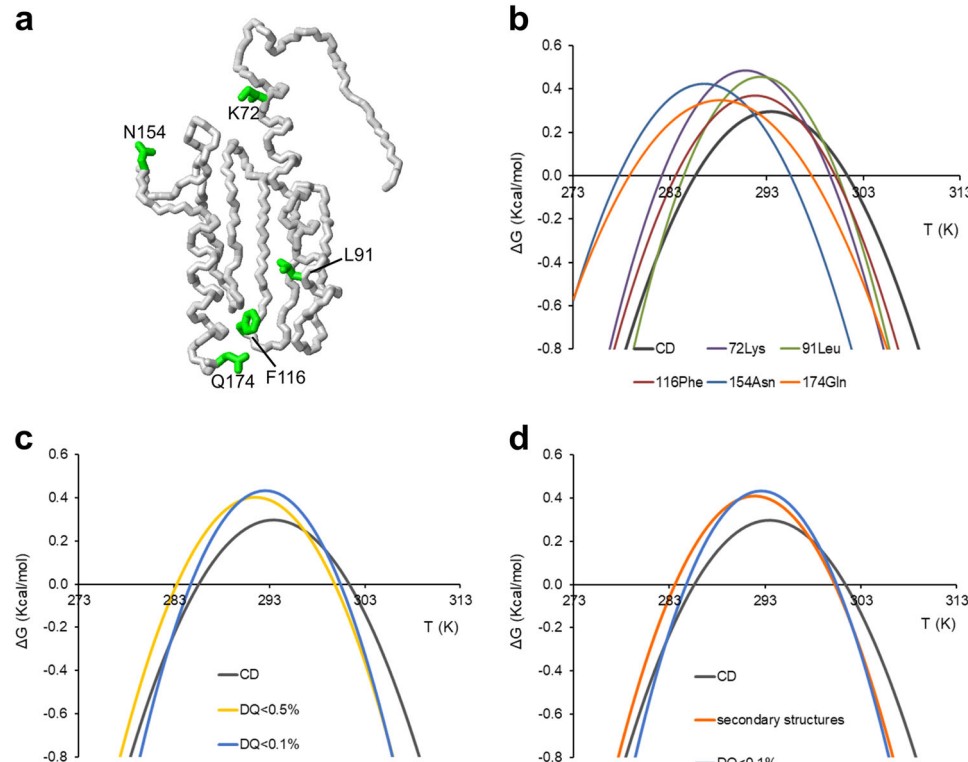

**Fig. 3 Comparison of stability curves calculated from CD and NMR data. a** Neon representation of Yfh1 (2fql) with side chains of the five residues corresponding to the stability curves from CD (dashed line), from four arbitrarily selected individual NMR resonances (K72, L91, F116 and N154) and the C-terminal Q174. **b** Stability curves from CD (grey) and NMR data for single residues: K72 (violet), L91 (green), F116 (brown), N154 (pale blue) and Q174 (orange). **c** Stability curves from CD (grey) and NMR data from interior residues (yellow for RAD < 0.5 and pale blue for RAD < 0.1). **d** Comparison of the stability curves from CD (grey), NMR selecting residues in secondary structure elements (orange) and NMR selecting interior residues (pale blue for RAD < 0.1). The model was generated by MOLMOL[34].

selecting ring-current shifted resonances. It will be interesting in the future to closely reconsider whether the differences observed among the parameters obtained for individual resonances and their differences with the CD curve hold valuable structural information.

Could have we chosen completely different selection criteria? Certainly. We could for instance have used the set of cross-peaks corresponding to the slower solvent exchange amide protons. This approach would be based on an NMR procedure established in the '90s to get the $\Delta G$ of unfolding by measuring the kinetic rates of H/D exchange of amide protons[26,27]. $\Delta G$ would then be obtained from the set of amide protons showing slower H/D solvent exchange. While the two approaches should in principle yield similar results because selecting in different ways protected protons, the method presented here has the advantage of its simplicity and should not be affected by additional parameters such as pH and the individual residue acidity. If the kinetics of the exchange regime allowed, the approach could be coupled with the use of zz-exchange spectroscopy-based methods: these experiments are appropriate to quantitate transitions between different conformations of biomolecules for which isolated resonances are observed[28,29].

No problem is expected to arise if the protein required different 2D-based pulse sequences such as TROSY, since the approach should be easily extendible.

In conclusion, the possibility of reliably using 2D NMR for stability measurements opens entirely new avenues. It will allow to study the thermodynamics of protein unfolding in complex environments such as in-cell NMR or any other mesh of molecules, which would otherwise be unapproachable by other techniques.

## Methods

**Protein purification**. The two proteins were expressed in BL21(DE3) E. coli as previously described[8,30,31]. To obtain uniformly $^{15}N$-enriched untagged Yfh1, bacteria were grown in M9 using $^{15}N$-ammonium sulphate as the sole source of nitrogen until an OD of 0.6–0.8 and induced for 4 h at 37 °C with 0.5 mM IPTG. Purification involved two precipitation steps with $(NH_4)_2SO_4$, dialysis and anion exchange chromatography with Q-sepharose column using a NaCl gradient. After dialysis the protein was further purified by a chromatography with a Phenyl Sepharose column with a decreasing gradient of $(NH_4)_2SO_4$. CyaY was expressed as His, GST-tagged protein and obtained as selectively labelled on tyrosines by growing the cells in M9 medium supplemented with $^{14}N$-ammonium sulphate, $^{12}C$ glucose, 0.5 g of each unlabelled amino acid and 0.1 g of $^{15}N$-labelled tyrosine. Purification involved a first step with an affinity chromatography (with Ni-NTA agarose) followed by a dialysis with cleavage of the His,GST-tag by TEV protease. The protein was further purified by gel filtration chromatography on a 16/60 Superdex G75 column. Purity of the recombinant proteins was checked by sodium dodecyl sulfate polyacrylamide gel electrophoresis after each step of the purification.

**NMR experiments**. Two-dimensional (2D) NMR HSQC experiments were acquired on a Bruker AVANCE spectrometer operating at 700 MHz proton frequency. Measurements were carried out in 10 mM Hepes at pH 7.5 using 0.1 mM $^{15}N$-labelled Yfh1 mixed with 0.1 mM CyaY with selectively $^{15}N$-labelled tyrosine 69. No additional salt was added on purpose to allow observation of the cold denaturation process[15,18]. Spectra were recorded from 5 to 40 °C with intervals of 2.5 °C and using the Watergate water suppression sequence[32]. Eight scans were accumulated for each increment, for a total of 240 increments. HSQCs of the individual proteins were also independently recorded to ascertain that both proteins were correctly folded with the resonances in the expected position[15,30]. Spectra were processed with NMRPipe and analysed with the CCPNMR software. Gaussian (LB−15 and GB 0.1), and cosine window functions were applied for the

direct and indirect dimension, respectively. The data were zero-filled twice in both dimensions. Spectral assignment of Yfh1 was based on the BMRB deposition entry 19991.

**CD experiments**. Measurements were performed using a 10 μM protein concentration in 10 mM Hepes at pH 7.5. Baseline was corrected by subtracting the buffer spectrum. Thermal unfolding curves were generated by monitoring the ellipticity at 222 nm in a Jasco J-815 CD spectropolarimeter equipped with a Jasco CDF-4265/15 Peltier unit. One millimetre path length cells (Hellma) were used, at a heating rate of 1 °C/min in the temperature range 2–70 °C, D.I.T of 4 s and bandwidth 1 nm.

**Volume corrections**. Volumes were calculated by summation of intensities in a set box using the CCPNMR software. Volume corrections were performed dividing each resonance of Yfh1 at a given temperature for the resonance of CyaY Tyr69 amide at the same temperature. This procedure is conceptually similar to what is now routinely done in metabolomics[33], another field in which populations are important. Our purpose was however slightly different: in metabolomics accurate quantification of the populations is needed and thus the nature of the reference resonance is crucial. In our case we were instead interested in getting rid of instrumental effects.

Two examples of fitting of folded populations, relative to stability curves of Figs. 2 and 3, are reported in Supplementary Fig. 2.

**Safe criteria for selecting the best set of amides of Yfh1**. Yfh1 contains 114 backbone amide protons out of its 123 residues (numbering of the mature protein starts at residue Glu53 and ends at Gln174). Of these, the first 23 residues are in an unstructured N-terminal tail that contains the region for mitochondrial import and processing, leading to 91 resonances in the globular domain. Sixty-six residues (75%) have non-overlapping and isolated resonances that allow easily detectable and reliable volume calculation. Notice that most of the excluded overlapping resonances corresponded to disordered regions or, as proven before, to a partially unfolded conformation in equilibrium with the folded one in a slow exchange regime at room temperature[15,18]. Shift of the equilibrium towards a fully folded form can easily be achieved increasing the ionic strength but, under these conditions, no cold denaturation is observed.

The parameter RAD was used taking the parameters from the software Pops (http://mathbio.nimr.mrc.ac.uk/~ffranca/POPS) and SADIC (http://www.sbl.unisi.it/prococoa/). Residues involved in secondary structures were evaluated according to the DSSP programme (https://swift.cmbi.umcn.nl/gv/dssp/). This software is all freely available. Our analysis resulted in 35 residues in secondary structure elements (15 in alpha helices, 20 in beta sheets), 39 residues having RAD < 0.5, 37 with RAD < 0.4, 33 with RAD < 0.3, 24 with RAD < 0.2 and 11 having RAD < 0.1 (Supplementary Table 1).

## Data availability
The data that support the findings of this study are available from the corresponding author on reasonable request.

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

## Acknowledgements
The research was supported by UK Dementia Research Institute (RE1 3556), which is funded by the Medical Research Council, Alzheimer's Society and Alzheimer's Research UK. We also thankfully acknowledge the Francis Crick Institute for provision of access to the MRC Biomedical NMR Centre. The Francis Crick Institute receives its core funding from Cancer Research UK (FC001029), the UK Medical Research Council (FC001029) and the Wellcome Trust (FC001029). We thank Geoff Kelly and Tom Frenkiel of the MRC Biomedical NMR Centre for helpful discussions and technical support, Neri

Niccolai and Franca Fraternali for help with their software SADIC and PopS, respectively. We also acknowledge the use of the NMR spectrometers at the Randall unit of King's College London.

## Author contributions

All of the authors contributed to the research design and data analyses. R.P. and O.B. performed most of the experiments described here. R.P., C.A. and S.R.M. performed the extraction of the thermodynamic data. A.P. supported the work financially, provided the protein material, and helped writing the manuscript. P.A.T. coordinated the work and wrote the manuscript.

## Competing interests

The authors declare no competing interests.
