## [Peer Review File · Communications Chemistry]

Reviewers' comments:

Reviewer #1 (Remarks to the Author):

This is an interesting article, presenting the new approach for observation of protein folding employing peak volumes in 2D NMR spectra. The main achievement is the successful application of internal standard for normalization of peak volumes.

The article is well written and should be published.

However, for the sake of clarity Authors should discuss the role of chemical exchange of amide protons with water. The rates of the exchange depend on many factors, like pH, temperature, and most importantly folding state.

Reviewer #2 (Remarks to the Author):

In the current manuscript, Puglisi et al. propose and evaluate a 2D NMR-based method to get thermodynamic parameters for proteins. The importance and advantage of the proposed method lies in their applicability to proteins in complex experimental conditions, or even in-cell, which is not possible for the current techniques used for in vitro protein stability studies. Protein unfolding is followed by 2D ¹H,¹⁵N-HSQC spectra, where a cross-peak is observed for the amide group of every residue in the protein. Yfh1, a protein, which suffers cold and heat denaturation, is used as model to evaluate the method. Interestingly, the authors find that the "goodness" of the determined thermodynamics (or coincidence with those obtained by CD), depends on the set of amino acids used for fitting the changes in cross-peak volumes upon temperature variation. To get reliable measurements of cross-peak volumes, a protein labelled in a single residue is incorporated as a reference. A procedure to select the residues, which are representative of the global protein unfolding, is described. On the whole, I find the manuscript is well written, the experimental work looks sound and the aim and results are of relevance. Hence, the manuscript merits to be published at the journal Communications Chemistry. However, I consider that there is a few points to be clarified prior publication.

1. My main point refers to the criteria to select the set of amide cross-peaks, which is the best to reflect the global protein unfolding. The proposed criterion seems to be reliable, but I would like to know if the authors have considered alternative criteria:

1.1. I wonder whether a criterion based only in the Solvent accessible surface area (SASA, which is listed in Table S1) could provide reliable thermodynamics parameters.

1.2. Other selection criterion could be to use the set of cross-peaks corresponding to the slowest solvent exchange amide protons. This would be based on the NMR procedure established in the 1990's to get the free energy of unfolding of a protein (ΔG), which consists in measuring the kinetic rates of H/D exchange of amide protons (Woodward 1994, Curr Opin Struct Biol 4:112–116; Englander et al. 1996, Curr Opin Struct Biol 6:18–23). Then, the free energy of protein unfolding (ΔG) is obtained from the set of amide protons showing the slowest H/D solvent exchange. It would be interesting to identify the slowest solvent exchange amide protons of protein Yfh1, and examined how the thermodynamics parameters obtained from them compared to those derived using the parameter RAD (relative accessibility and depth).

2. A minor point is that it would be convenient to include the definitions for "Depth", "SASA", "RA" and "DQ" at the title of Table S1.

Reviewer #3 (Remarks to the Author):

See attached.

Review of “Thermodynamics of protein unfolding in complex environments: using 2D NMR to measure protein stability curves” by Puglisi et al.

In this manuscript the authors present an extension of their previously published work on monitoring protein unfolding using the protein Yfh1, which can undergo both cold and heat denaturation in an experimentally tractable temperature range in low ionic strength buffers, and which populates a partially unfolded state to a substantial population fraction even at the temperature that most favours the folded form. The authors propose a method for using multidimensional NMR spectroscopy to analyse such systems on the grounds that this is more likely to be applicable than other methods, such as CD, to proteins studied in complex mixtures such as *in cellulo*. The main conclusion of the work presented is that the careful selection of reporter signals is important to produce results that agree with those from e.g. CD and 1D NMR studies performed under similar *in vitro* conditions. A protocol for selecting suitable reported signals is described based on a combination of algorithms that evaluate how buried each residue is in the globular structure.

The work points in an interesting direction but does not deliver on the title since no “complex environments” are tested. Together with rather imprecise use of terminology, missing experimental details, a very limited presentation of the experimental results and some issues that warrant further consideration, the manuscript should be revised.

The method depends on the accurate quantification of the relative populations of the folded and unfolded states at each experimental condition as indicated by the NMR signal intensities of the selected resonances (atoms or pairs of atoms in this case). As far as can be judged from the text but not made explicit, the authors are relying on quantifying the signal intensity from one (the folded) state since the signals from the unfolded state are too overlapped to be reliably quantified. This perhaps explains why the authors expend a substantial proportion of the paper on a proposed method for normalisation of signal intensities by **inclusion of an internal standard** – in this case another protein of similar size and shape. Some details are missing or need clarification here and some important considerations are not fully explored:

- The details of how the NMR data were recorded (number of repeat spectra at each temperature, pulse program used, values of key delays such as the recycle delay, etc.), processed (window functions, zerofilling, etc.) and integrated (by summation in a set box or by fitting an approximate or an accurate model of the lineshapes?) are incomplete.
- The incorporation of an internal standard is intended to allow correction for a number of factors that will affect the measured intensities. While the choice of a signal from a similar sized protein may compensate for differences in relaxation rate to an extent, it will not account for differences in ^{15}N offset (the authors chose a reference peak near the centre of their ^{15}N sweep width) or differences in coupling constant as claimed.
- The choice of a single site in the internal standard seems to be at odds with their own evidence that it's necessary to examine multiple sites and that the residues that are the most deeply buried are unexpectedly unrepresentative.
- Since even in some cases in the slow exchange regime the effective R1 & R2 for the signals of one state are a combination of those for both states, which are likely to be very different for the folded and unfolded states, and are also affected by the rate of exchange between states, a fully folded standard may not be an adequate model.

Ultimately, the authors show that the normalization to the internal standard has a negligible effect on the fitted stability curves. This is despite a more than 1.5 fold difference in the relative signal intensity from the standard across the temperature range (Fig S1). The insensitivity of the

thermodynamic model to this difference in input is surely worthy of more comment and possibly a concern for the method as a whole. It may be that, if the kinetics of the exchange regime allow, a zz-exchange spectroscopy based method would solve many of the problems that the authors seek to address.

Given the history of work in this field, it is understandable that the authors focus on finding a protocol for selecting resonances that will produce data that closely match those from CD and thus find that they need to restrict their analysis of the NMR data to a set of the most buried residues. The disagreement with CD for non-core residues is presented as a weakness, whereas it is perhaps a strength of the technique, since it provides a richer data set that reports on processes that occur in addition to the cooperative melting of the protein core that other techniques would struggle to access. Nonetheless, the description of their chosen **RAD algorithm** lacks some of the details that would enable others to reproduce it and there are some unexpected outcomes that warrant more attention.

- The RAD algorithm proposed combines the results of SADIC, which produces a score on a scale 0-2 describing how buried a residue is in the protein structure, and PopS, which can provide estimates of solvent accessible surface area either on a per-atom or a per-residue basis as well as a the fraction of an atom or residue's surface area that is accessible. It is not clear from the description in the paper, what settings were used in each of the programs. Was SADIC evaluating depth based on C α or other atom positions? Was PopS used in residue or atom mode and if atom mode, which atom was it set to report on. For the RAD calculation, it should be clarified that Q(SASA) of the residue or atom was used in the calculation – the term relative accessibility (RA) is not used in the PopS publications or documentation.
- If PopS is used in residue mode, then SADIC and PopS should provide similar discrimination of a residue's position in the structure for such a relatively small protein which has very few completely buried residues. The question therefore arises as to what the product of the two parameters achieves that either one alone can not.
- There seem to be some inconsistencies between the main text and the results presented in Table 1 and Figure 2. The authors seem to use DQ and RAD interchangeably.

The confidence intervals for the fitted thermodynamic parameters presented in table 1 have surprisingly narrow confidence intervals in most cases. Since the method of **fitting and error estimation** are not described in the methods and no fits to the NMR intensities are shown (cf <https://pubs.acs.org/doi/10.1021/ja0714538>), it is difficult to have confidence that the claimed precision of the fits is reasonable. It would be informative to see the fits to the NMR intensities plotted along with a more complete description of the error analysis. The stability curve for residues with RAD < 0.1 is claimed to display "excellent" agreement with the CD derived data, but that is far from obvious either by visual inspection, or comparison of the extracted numerical parameters. Table 1 also seems to be missing one of the ΔH parameters ($\Delta H_c?$).

In "complex mixtures" such as in cells, the increased viscosity often mandates the use of pulse sequences other than the HSQC. It would be informative to know whether similar considerations are required, and performance can be expected if, for example, a TROSY pulse sequence were employed.

Some specific suggestions:

The sentence “We thus argued that an informed but not so strict selection of the reference would not be necessary.” is unclear.

Where “114 amide protons” are mentioned, does this mean backbone amides or does include amide side chains?

In Fig 2, the colours should be explained in the main figure legend as the information presented on each sub plot is so small as to be illegible.

In the Fig2 legend – were the residues really randomly selected, or arbitrarily?

We wish to thank the reviewers for the constructive criticism to our work.

There was perhaps a slight misunderstanding with reviewer #3. Although most criticism is right and helpful, we have the impression that this reviewer did not take into account that the manuscript was conceived as a communication, i.e. a concise and compact report on a subject of interest, urgency and wide interest, not a full paper. This is why the format was so compact and without subparagraphs. In hindsight we realized that we probably did limit the length of the text excessively. We have done our best to integrate the information and to answer all reviewers' comments as detailed below. We have used bold face times to indicate our answers to facilitate comprehension. We also highlighted in green the changes introduced in the text. We hope that the reviewers will find our answers satisfactory.

[Reviewer #1 (Remarks to the Author):

This is an interesting article, presenting the new approach for observation of protein folding employing peak volumes in 2D NMR spectra. The main achievement is the successful application of internal standard for normalization of peak volumes.

The article is well written and should be published.

However, for the sake of clarity Authors should discuss the role of chemical exchange of amide protons with water. The rates of the exchange depend on many factors, like pH, temperature, and most importantly folding state.]

We thank the reviewer for the kind words of appreciation of our work.

The reviewer is right about the importance of discussing the role of chemical exchange. In fact, a discussion was present in an earlier version of the manuscript but was later on omitted. We have now reintroduced the text back both in the introduction and as a discussion.

[Reviewer #2 (Remarks to the Author):

In the current manuscript, Puglisi et al. propose and evaluate a 2D NMR-based method to get thermodynamic parameters for proteins. The importance and advantage of the proposed method lies in their applicability to proteins in complex experimental conditions, or even in-cell, which is not possible for the current techniques used for in vitro protein stability studies. Protein unfolding is followed by 2D ¹H,¹⁵N-HSQC spectra, where a cross-peak is observed for the amide group of every residue in the protein. Yfh1, a protein, which suffers cold and heat denaturation, is used as model to evaluate the method. Interestingly, the authors find that the 1C;goodness 1D; of the determined thermodynamics (or coincidence with those obtained by CD), depends on the set of amino acids used for fitting the changes in cross-peak volumes upon temperature variation. To get reliable measurements of cross-peak volumes, a protein labelled in a single residue is incorporated as a

reference. A procedure to select the residues, which are representative of the global protein unfolding, is described. On the whole, I find the manuscript is well written, the experimental work looks sound and the aim and results are of relevance. Hence, the manuscript merits to be published at the journal Communications Chemistry.

We thank the reviewer for the appreciation of our work.

However, I consider that there is a few points to be clarified prior publication.

1. My main point refers to the criteria to select the set of amide cross-peaks, which is the best to reflect the global protein unfolding. The proposed criterion seems to be reliable, but I would like to know if the authors have considered alternative criteria:

1. This is a fair comment which we had indeed carefully evaluated. As mentioned in the manuscript, we explored the alternative criterion of using only residues belonging to traits of secondary structure. However, it is important to take into account that the most important step in comparing NMR and CD data is averaging NMR data over a large number of residues. The averaging in CD and NMR is very different (Kruschel D, Zagrovic B. Conformational averaging in structural biology: issues, challenges and computational solutions. *Mol Biosyst.* 2009;5(12):1606-1616. doi:10.1039/b917186j) and this is an important aspect of the present work.

1.1. I wonder whether a criterion based only in the Solvent accessible surface area (SASA, which is listed in Table S1) could provide reliable thermodynamics parameters.

1.1 This is a very appropriate comment. We did try simpler criteria, such as using only SASA values to select buried residues, but these numbers are calculated on the whole residue and not only on the amides. This turned to be decidedly inappropriate, because it does not take into account the actual position of the amide hydrogen. It may happen that most of the residue atoms are buried whereas the amide hydrogen is accessible to water. This is an interesting problem that has a general validity. It has been considered by several people and we are trying to merge together different approaches.

1.2. Other selection criterion could be to use the set of cross-peaks corresponding to the slowest solvent exchange amide protons. This would be based on the NMR procedure established in the 1990 19;s to get the free energy of unfolding of a protein (ΔG), which consists in measuring the kinetic rates of H/D exchange of amide protons (Woodward 1994, *Curr Opin Struct Biol* 4:112 13;116; Englander et al. 1996, *Curr Opin Struct Biol* 6:18 13;23). Then, the free energy of protein unfolding (ΔG) is obtained from the set of amide protons showing the slowest H/D solvent exchange. It would be interesting to identify the slowest solvent exchange amide protons of protein Yfh1, and examined how the thermodynamics parameters obtained from them compared to those derived using the parameter RAD (relative accessibility and depth).

Taking into account the exchange of amide protons is indeed very relevant. We had originally planned to use precisely this criterion for the choice of the residues to average. However, in the end we decided to postpone the idea to a later development because we wanted to find an approach of the widest possible applicability, without burdening the determination of thermodynamic parameters in complex environments

with a long experimental work. But the suggestion is excellent and we have added a couple of sentences that resonate directly with the reviewer's suggestion.

2. A minor point is that it would be convenient to include the definitions for 1C;Depth 1D;, 1C;SASA 1D;, 1C;RA 1D; and 1C;DQ 1D; at the title of Table S1.]

2. Of course. We have included the definitions in Table S1.

[Reviewer #3]

[Review of "Thermodynamics of protein unfolding in complex environments: using 2D NMR to measure protein stability curves" by Puglisi et al.

In this manuscript the authors present an extension of their previously published work on monitoring protein unfolding using the protein Yfh1, which can undergo both cold and heat denaturation in an experimentally tractable temperature range in low ionic strength buffers, and which populates a partially unfolded state to a substantial population fraction even at the temperature that most favours the folded form. The authors propose a method for using multidimensional NMR spectroscopy to analyse such systems on the grounds that this is more likely to be applicable than other methods, such as CD, to proteins studied in complex mixtures such as *in cellulo*. The main conclusion of the work presented is that the careful selection of reporter signals is important to produce results that agree with those from e.g. CD and 1D NMR studies performed under similar *in vitro* conditions. A protocol for selecting suitable reported signals is described based on a combination of algorithms that evaluate how buried each residue is in the globular structure. The work points in an interesting direction but does not deliver on the title since no "complex environments" are tested. Together with rather imprecise use of terminology, missing experimental details, a very limited presentation of the experimental results and some issues that warrant further consideration, the manuscript should be revised.

The method depends on the accurate quantification of the relative populations of the folded and unfolded states at each experimental condition as indicated by the NMR signal intensities of the selected resonances (atoms or pairs of atoms in this case). As far as can be judged from the text but not made explicit, the authors are relying on quantifying the signal intensity from one (the folded) state since the signals from the unfolded state are too overlapped to be reliably quantified. This perhaps explains why the authors expend a substantial proportion of the paper on a proposed method for normalisation of signal intensities by **inclusion of an internal standard** – in this case another protein of similar size and shape. Some details are missing or need clarification here and some important considerations are not fully explored:]

We thank the reviewer for the extensive analysis of our work, which stimulated many new ideas and, hopefully, led to an improved manuscript.

Below are our answers to the specific points:

[The details of how the NMR data were recorded (number of repeat spectra at each temperature, pulse program used, values of key delays such as the recycle delay, etc.), processed (window functions, zerofilling, etc.) and integrated (by summation in a set box or by fitting an approximate or an accurate model of the lineshapes?) are incomplete.]

The reviewer is right. However, this work was originally meant as a short communication (in the classical sense), intended as a concise and compact report, not as a full paper. This for us is only the starting point for a more extensive and focused investigation. This is why the format is so compact, although on hindsight we realized

that in the end we probably did limit the length of the text excessively. We apologise for this and thank the reviewer for the suggestions. We have now tried to add as much information as possible to improve comprehension and reproducibility.

[The incorporation of an internal standard is intended to allow correction for a number of factors that will affect the measured intensities. While the choice of a signal from a similar sized protein may compensate for differences in relaxation rate to an extent, it will not account for differences in ^{15}N offset (the authors chose a reference peak near the centre of their ^{15}N sweep width) or differences in coupling constant as claimed.]

The reviewer is certainly right, but it was necessary to reach a compromise among different requirements. The reason we chose the amide hydrogen of Tyr69 of CyaY is that its resonance is well apart from those of Yfh1 and this residue is the only Tyr in the whole CyaY, facilitating selective labelling and recognition.

[The choice of a single site in the internal standard seems to be at odds with their own evidence that it's necessary to examine multiple sites and that the residues that are the most deeply buried are unexpectedly unrepresentative.]

The goal of this work was not to find the “ultimate solution” to correct ALL resonances but rather to propose a simple method to use NMR data in a way consistent with CD spectroscopy, the most used spectroscopic method to extract thermodynamic parameters on protein stability. Yes. For a full study, we would repeat the analysis with 2-3 residues. This is potentially possible.

[Since even in some cases in the slow exchange regime the effective R_1 & R_2 for the signals of one state are a combination of those for both states, which are likely to be very different for the folded and unfolded states, and are also affected by the rate of exchange between states, a fully folded standard may not be an adequate model.

Ultimately, the authors show that the normalization to the internal standard has a negligible effect on the fitted stability curves. This is despite a more than 1.5 fold difference in the relative signal intensity from the standard across the temperature range (Fig S1). The insensitivity of the thermodynamic model to this difference in input is surely worthy of more comment and possibly a concern for the method as a whole.

It may be that, if the kinetics of the exchange regime allow, a zz exchange spectroscopy based method would solve many of the problems that the authors seek to address. Given the history of work in this field, it is understandable that the authors focus on finding a protocol for selecting resonances that will produce data that closely match those from CD and thus find that they need to restrict their analysis of the NMR data to a set of the most buried residues. The disagreement with CD for non-core residues is presented as a weakness, whereas it is perhaps a strength of the technique, since it provides a richer data set that reports on processes that occur in addition to the cooperative melting of the protein core that other techniques would struggle to access. Nonetheless, the description of their chosen **RAD algorithm** lacks some of the details that would enable others to reproduce it and there are some unexpected outcomes that warrant more attention.]

It would be ideal to be able to measure the population of both folded and unfolded states accurately. However, in nearly all situations (and with nearly ALL methods) this is not possible. It is true that in our case the exceptional features of the protein might allow, in principle, the measurements of the intensity of a few resonances attributable to the unfolded form. However, most of these resonances are overlapping with other resonances and thus not usable.

We were ourselves surprised of the relatively small influence of normalization and had to repeat the analysis to convince ourselves. In our opinion, the results show that for Yfh1 averaging is far more important than normalisation to obtain a good comparison between CD and NMR data. Nonetheless, our proposal of an internal standard retains its validity in a general case.

As for zz exchange spectroscopy, this approach would be ideal to minimise problems. We are fully aware of the potentiality of this method, also because one of the co-authors was co-author of the first paper on this subject (Oschkinat H, Pastore A, Pfändler P, Bodenhausen G. (1986) Two-dimensional correlation of directly and remotely connected transitions by z-filtered COSY *Journal of Magnetic Resonance* 69: 559-566). However, as stated previously, we sought a simple method available to most laboratories studying protein stability and, to the best of our knowledge, the use of zz exchange spectroscopy is not so widespread. It also needs a specific exchange regime.

We fully agree that it is in no way right to present the disagreement with CD for non-core residues as a weakness. Our hope in designing this study was to find a set of residues (likely those of the protein core) that would yield thermodynamic parameters consistent with those given by CD spectroscopy and then explore the whole range of residues to gather information on the mechanisms of unfolding. Unfortunately, we found that parameters derived from single residues are utterly unreliable to represent the overall process of unfolding even though, as the reviewer correctly says, their different behaviour could contain interesting information which is nevertheless difficult to decode at the present. Thus, in our follow-up studies we plan to compare averaged data with those derived from selected categories of residues. We introduced a couple of sentences to clarify this point.

We also agree that the description of the RAD parameter was missing important details and have done our best to improve it.

[The RAD algorithm proposed combines the results of SADIC, which produces a score on a scale 0-2 describing how buried a residue is in the protein structure, and PopS, which can provide estimates of solvent accessible surface area either on a per-atom or a per-residue basis as well as the fraction of an atom or residue's surface area that is accessible. It is not clear from the description in the paper, what settings were used in each of the programs. Was SADIC evaluating depth based on C α or other atom positions? Was PopS used in residue or atom mode and if atom mode, which atom was it set to report on. For the RAD calculation, it should be clarified that Q(SASA) of the residue or atom was used in the calculation – the term relative accessibility (RA) is not used in the PopS publications or documentation.]

As clarified previously, we did not provide too many details only within the idea of a short communication. This said, we have now added full details and corrected some minor incorrections. In a nutshell: both SADIC and PopS evaluate the state of amide groups but using two quite different approaches (see below).

[If PopS is used in residue mode, then SADIC and PopS should provide similar discrimination of a residue's position in the structure for such a relatively small protein which has very few completely buried residues. The question therefore arises as to what the product of the two parameters achieves that either one alone can not.]

The two programs used are not equivalent because they are based on completely different premises. SADIC can tell how far inside the structure an atom is located.

PopS, when used in the atomic mode, can tell how much the same atom is solvent accessible. We believe that the combination of the two methods is more effective to tell, in an unbiased and objective way, whether an amide group is buried, i.e. telling at the same time whether the atom is deeply inside the structure AND not solvent exposed.

[There seem to be some inconsistencies between the main text and the results presented in Table 1 and Figure 2. The authors seem to use DQ and RAD interchangeably. The confidence intervals for the fitted thermodynamic parameters presented in table 1 have surprisingly narrow confidence intervals in most cases. Since the method of **fitting and error estimation** are not described in the methods and no fits to the NMR intensities are shown (cf <https://pubs.acs.org/doi/10.1021/ja0714538>), it is difficult to have confidence that the claimed precision of the fits is reasonable. It would be informative to see the fits to the NMR intensities plotted along with a more complete description of the error analysis. The stability curve for residues with RAD < 0.1 is claimed to display “excellent” agreement with the CD derived data, but that is far from obvious either by visual inspection, or comparison of the extracted numerical parameters. Table 1 also seems to be missing one of the $\otimes\text{H}$ parameters ($\otimes\text{H}_c?$). In “complex mixtures” such as in cells, the increased viscosity often mandates the use of pulse sequences other than the HSQC. It would be informative to know whether similar considerations are required, and performance can be expected if, for example, a TROSY pulse sequence were employed.]

We apologise. The inconsistency arose from a change between two of the last versions of the manuscript. We first used DQ and then changed to RAD to avoid confusion. We have now corrected it.

About the error estimation we completely agree. This is however a very recurrent problem. Errors evaluated on fitting are usually ridiculously low and cannot be trusted. On the other hand, it is difficult to follow accurately error propagation of data that come from averaging and with so many steps of processing. This is why some times it may be more appropriate not to provide the error at all.

As for the excellent agreement, it is extremely difficult to find in the literature two measurements of T_m made by the same technique (e.g. CD) on the same protein that are as close as 300.4 (RAD<0.1) and 301.3 (CD), as reported in Table 1. A difference of 1-2 degrees is well within the experimental error of the measurements. It is only 0.9 degrees in our case. This is below the error that we found averaging the data among more than 10 years of measurements on entirely different preparations of the same protein.

We cannot see $\otimes\text{H}$ parameters in Table 1. Perhaps some symbol was not exported faithfully from word to pdf.

We agree that actual “complex mixtures” may eventually require different pulse sequences. However, the logic should in principle be the same. We have added a sentence in the text to comment this.

Some specific suggestions:

[The sentence “We thus argued that an informed but not so strict selection of the reference would not be necessary.” is unclear.]

The reviewer is right, we have deleted the sentence.

[Where “114 amide protons” are mentioned, does this mean backbone amides or does

include amide side chains?]

Only backbone amides. We have now clarified this point in the text (p. 9).

[In Fig 2, the colours should be explained in the main figure legend as the information presented on each sub plot is so small as to be illegible.]

We have now amended the figure legend.

[In the Fig. 2 legend – were the residues really randomly selected, or arbitrarily?]

If with randomly we mean through a random generation seed, no. We arbitrarily selected the residues.

REVIEWERS' COMMENTS:

Reviewer #2 (Remarks to the Author):

In my opinion, in the revised version of the manuscript Puglisi et al. have addressed adequately all the points raised by myself and the other reviewers on the original version. Hence, I find the manuscript suitable for publication.

Reviewer #3 (Remarks to the Author):

The revised manuscript is much improved. I still think the title over-promises, and it would be interesting to see the data being fitted in figs 2 & 3 (perhaps as supplementary figures) though.

Gaaino PDF Trial
www.gaaino.com